# COUNTERFACTUAL IMAGE GENERATION FOR ADVERSARIALLY ROBUST AND INTERPRETABLE CLASSIFIERS

## ABSTRACT

Neural Image Classifiers are effective but inherently hard to interpret and susceptible to adversarial attacks. Solutions to both problems exist, among others, in the form of counterfactual examples generation to enhance explainability or adversarially augment training datasets for improved robustness. However, existing methods exclusively address only one of the issues. We propose a unified framework leveraging image-to-image translation Generative Adversarial Networks (GANs) to produce counterfactual samples that highlight salient regions for interpretability and act as adversarial samples to augment the dataset for more robustness. This is achieved by combining the classifier and discriminator into a single model that attributes real images to their respective classes and flags generated images as "fake". We assess the method's effectiveness by evaluating (i) the produced explainability masks on a semantic segmentation task for concrete cracks and (ii) the model's resilience against the Projected Gradient Descent (PGD) attack on a fruit defects detection problem. Our produced saliency maps are highly descriptive, achieving competitive IoU values compared to classical segmentation models despite being trained exclusively on classification labels. Furthermore, the model exhibits improved robustness to adversarial attacks, and we show how the discriminator's "fakeness" value serves as an uncertainty measure of the predictions.

## 1 INTRODUCTION

In this study, we focus on Neural Networks (NN) for binary image classification, which have found applications in fields ranging from medical diagnosis (Marques et al., 2020; Albahar, 2019; Skouta et al., 2021) to structural health monitoring (Rao et al., 2021; Xu et al., 2019) and defect detection (Kwak et al., 2000; Tao et al., 2018; Drass et al., 2021). The remarkable precision, coupled with their computational efficiency during inference, enables seamless integration of NNs into existing systems and workflows, facilitating real-time feedback immediately after data acquisition, such as following a CT scan or while a drone captures images of concrete retaining walls to detect cracks.

Despite their capabilities, NNs have some shortcomings. They are susceptible to adversarial attacks that can deceive model predictions with subtle, human-imperceptible perturbations (Szegedy et al., 2013; Nguyen et al., 2015; Kurakin et al., 2018). Moreover, NNs typically lack interpretability, providing no rationale for their classifications. Efforts to improve interpretability have yielded techniques like Grad-CAM (Selvaraju et al., 2017) and RISE (Petsiuk et al., 2018), which produce attribution masks highlighting influential image regions. However, these masks are often blurry and lack precision (Adebayo et al., 2018; Ghorbani et al., 2019; Stalder et al., 2022; Riedel et al., 2022). Recent research has explored counterfactual-based explanations using GANs to address these limitations (Chang et al., 2018; Nemirovsky et al., 2020; Charachon et al., 2022; Mertes et al., 2022).

These attribution methods are typically implemented post-hoc, implying that the classifier is pretrained and remains unaltered during the counterfactual training process. In contrast, methods enhancing robustness train classifiers and GANs concurrently (Tsipras et al., 2018; Woods et al., 2019). By rigidly fixing the classifier's parameters, current explainability approaches forfeit the opportunity to train a more robust classifier. Our approach combines these methodologies, allowing for joint training of classifiers and GANs. This not only uses adversarial samples for interpretation but also creates classifiers resistant to minor, imperceptible image modifications. The result is a more robust classifier and GANs that generate relevant, visually coherent counterfactuals.

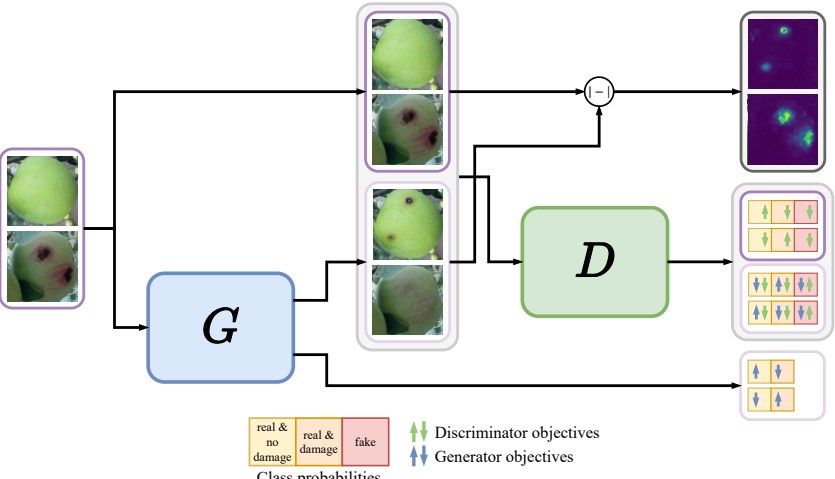

Figure 1: Overview of our Counterfactual Image Generation Framework. Input images from both classes are converted to class predictions and counterfactual samples by the generator $G$. The real and generated images are classified by the discriminator $D$ as real and belonging to class 0, real and belonging to class 1, or fake. Conversely, $G$ must deceive $D$ by producing realistic samples attributed to the opposite class by $D$. The absolute difference between real and generated images highlights salient regions.

Therefore, we introduce a unified framework that merges the generation of adversarial samples for enhanced robustness with counterfactual sample generation for improved interpretability. We extend the binary classification objective to a 3-class task, wherein the additional class signifies the likelihood that a sample has been adversarially modified, thus combining the discriminator and classifier into a single model. Conversely, the generator is responsible for image-to-image translation with a dual objective: to minimally alter the images such that they are classified into the opposing class by the discriminator and to ensure that these generated instances are indistinguishable from the original data distribution. This methodology has the benefits of (i) creating adversarial examples that augment the dataset, making the classification more robust against subtle perturbations, and (ii) creating pairs of original input images and their counterfactual versions whose absolute difference reveals the most salient regions employed by the classifier for making the predictions.

In summary, our contributions are:

- An end-to-end framework that merges adversarial robustness and explanations.

- Specialized architectures of our generator $G$ and discriminator $D$ specifically tailored to their respective objectives.

- Validation of our approach on two benchmark datasets: the CASC IFW database for fruit defects detection and the Concrete Crack Segmentation Dataset for structural health monitoring.

- Demonstrating improved robustness of our models against PGD attacks compared to conventional classifiers in addition to $D$ providing a reliable estimate of the model's predictive confidence.

- Qualitative and quantitative analysis showing that our method, trained on classification labels, significantly outperforms existing attribution techniques, such as GradCAM, in generating descriptive and localized saliency maps in addition to achieving an Intersection over Union (IoU) score that is only $12\%$ lower than models trained on pixel-level labels.

The rest of the paper is structured as follows: Section 2 summarizes related work, while our methodology is outlined in Section 3. In Section 4 we demonstrate the effectiveness of the approach with extensive empirical experiments. Finally, Section 5 provides a discussion of the results and Section 6 concludes the work.

## 2 RELATED WORK

**Adversarial Perturbations.** Neural Network image classifiers are prone to, often imperceptible, adversarial perturbations (Szegedy et al., 2013; Nguyen et al., 2015; Kurakin et al., 2018). The Fast Gradient Signed Method (FGSM) was introduced to generate adversarial examples using input gradients (Goodfellow et al., 2014). Building on this, Madry et al. (2017) proposed Projected Gradient Descent (PGD), an iterative variant of FGSM and considered a "universal" adversary among first-order methods.

Subsequent advancements include Learn2Perturb (Jeddi et al., 2020), which employs trainable noise distributions to perturb features at multiple layers while optimizing the classifier. Generative Adversarial Networks (GANs) have also been explored for crafting adversarial samples, where the generator aims to mislead the discriminator while preserving the visual similarity to the original input (Xiao et al., 2018; Zhang, 2019).

**Attribution Methods.** A different avenue for increasing trust in overparameterised black box NNs is to devise methodologies that explain their decision-making. In the realm of image classification, early techniques focused on visualizing features through the inversion of convolutional layers (Zeiler & Fergus, 2014; Mahendran & Vedaldi, 2015), while others employed weighted sums of final convolutional layer feature maps for saliency detection (Zhou et al., 2016). GradCAM advanced this by using backpropagation for architecture-agnostic saliency localization (Selvaraju et al., 2017). Extensions include GradCAM++ (Chattopadhay et al., 2018), Score-CAM (Wang et al., 2020), and Ablation-CAM, which forgoes gradients entirely (Ramaswamy et al., 2020). Gradient-free techniques like RISE (Petsiuk et al., 2018) employ random input masking and aggregation to compute saliency, whereas LIME (Ribeiro et al., 2016) uses a linear surrogate model to identify salient regions.

Combining both gradient-based and perturbation-based methods, Charachon et al. (2021a) utilize a linear path between the input and its adversarial counterpart to control the classifier's output variations. Fong & Vedaldi (2017) introduce gradient-based perturbations to identify and blur critical regions, later refined into Extremal Perturbations with controlled smoothness and area constraints (Fong et al., 2019). Generative models have also been employed for this purpose. Chang et al. (2018) generate counterfactual images using generative in-fills conditioned on pixel-wise dropout masks optimized through the Concrete distribution. Narayanaswamy et al. (2020) leverage a CycleGAN alongside a validation classifier to translate images between classes. A dual-generator framework is proposed by Charachon et al. (2021b) to contrast salient regions in classifier decisions, later refined with a discriminator to obviate the need for a reconstruction generator (Charachon et al., 2022). These frameworks use a pre-trained, static classifier, as well as different generators for translating images from one domain to another.

**Combining Adversarial and Attribution methods.** Tsipras et al. (2018) noted that non-minimal adversarial examples contained salient features when networks were adversarially trained, thus showing that perturbation could improve robustness and be used as explanation. Similarly, Woods et al. (2019) create adversarial perturbations of images subject to a Lipschitz constraint, improving the classifier's robustness and creating adversarial examples with discernible features for non-linear explanation mechanisms.

To the best of our knowledge, we are the first to explore the avenue of counterfactual image generation for achieving two critical goals: creating importance maps to identify the most salient regions in images and boosting the classifiers' robustness against adversarial attacks and noise injection. Previous works operate in a post-hoc manner, generating counterfactuals based on a pre-trained, static classifier. This approach limits the potential for the classifier to improve in terms of robustness, as it remains unaltered during the counterfactual training process and thus remains vulnerable to subtle perturbations that flip its predicted labels. In contrast, our method trains the generator and a combined discriminator-classifier model simultaneously. This end-to-end approach improves the classifier's interpretability by generating counterfactual samples and bolsters its robustness against adversarial perturbations.

## 3 METHODOLOGY

This study introduces a method that uses GANs to simultaneously improve the interpretability and robustness of binary image classifiers. We reformulate the original binary classification problem into a three-class task, thereby unifying the classifier and discriminator into a single model $D$. This augmented model classifies samples as either undamaged real images, damaged real images, or generated (fake) images.

The generator $G$ is tasked with creating counterfactual images; modified instances that are attributed to the opposite class when evaluated by $D$. As $D$ improves its discriminative capabilities throughout training, $G$ adaptively produces increasingly coherent and subtle counterfactuals.

These counterfactual images serve two essential roles: (i) they function as data augmentations, enhancing model robustness by incorporating subtle adversarial perturbations into the training set, and (ii) the absolute difference between original and counterfactual images highlights salient regions used by $D$ for classification, thereby improving model interpretability. The entire methodology is visually represented in Figure 1.

### 3.1 MODEL ARCHITECTURE

#### 3.1.1 GENERATOR

The generator $G$ performs image-to-image translation for transforming an input image $x$ into a counterfactual example $\hat{x} = G(x)$ that is misclassified by $D$: not only should $D$ be unable to detect that the counterfactual was generated by $G$, it should also attribute it to the wrong class. We employ a UNet (Ronneberger et al., 2015) architecture based on convolutional layers, denoted CNN UNet, as well as Swin Transformer blocks, named Swin UNet (Cao et al., 2022; Fan et al., 2022). To convert these networks into generative models, we adopt dropout strategies suggested by Isola et al. (2017) and introduce Gaussian noise at each upsampling layer for added variability.

In addition to the above, $G$ is equipped with an auxiliary classification objective. We implement this by adding a secondary branch after weighted average pooling from each upsampling block. This branch, comprised of a fully connected network, predicts the class label of the input image. The pooling operation itself is realized through a two-branch scoring and weighting mechanism (Yang et al., 2022). More detail about the modified UNet architecture is provided in Appendix A.1.

#### 3.1.2 DISCRIMINATOR

The discriminator, denoted as $D$, is tasked with both discriminating real from generated images and determining the class to which the real images belong. To achieve this, two additional output dimensions are introduced, extending the traditional *real/fake* scalar with values for *undamaged* and *damaged*. These values are trained using categorical cross-entropy and are therefore interdependent. Integrating the discriminator and classifier into a unified architecture creates a model whose gradients inform the generator to create realistic counterfactual images. Our experimentation explores various backbones for the discriminator, including ResNets and Swin Transformers of varying depths and a hybrid ensemble of both architecture types, combining a ResNet3 and a Swin Tiny Transformer into a single model. We will hereafter be referring to the latter as "Hybrid" discriminator.

### 3.2 TRAINING PROCESS

Consider $x$ to be an instance from the space of real images $\mathcal{X}$, which is partitioned into subsets $\mathcal{X}_y \subset \mathcal{X}$, each corresponding to a class label $y \in \{0, 1\}$. Our generator $G$ is a function $G : \mathcal{X} \to \mathcal{X} \times \mathbb{R}^2$, where $G(x) = (\hat{x}, \hat{y})$ includes both the counterfactual image $\hat{x}$ and a class probability vector $\hat{y} = [p_0, p_1]$. For notational convenience, we introduce $G_{img}(x) = \hat{x}$ and $G_{cls}(x) = \hat{y}$ as the image and class prediction branches, respectively. Our discriminator-classifier $D$ is characterized by the function $D : \mathcal{X} \to \mathbb{R}^3$, producing a tripartite output $D(x) = [p_0, p_1, p_{\text{fake}}]$ where each component represents the probability of $x$ being (i) real and from class 0 (undamaged), (ii) real and from class 1 (damaged) or generated (fake).

### 3.2.1 Loss functions

The discriminator $D$ is optimized through a categorical cross-entropy loss, which forces it to correctly classify real images while identifying those artificially generated by $G_{img}$.

$$\mathcal{L}_D = -\mathbb{E}_{x \sim \mathcal{X}_0} \left[ \log D(x)_0 \right] - \mathbb{E}_{x \sim \mathcal{X}_1} \left[ \log D(x)_1 \right] - \mathbb{E}_{x \sim \mathcal{X}} \left[ \log D(G_{img}(x))_{\text{fake}} \right] \tag{1}$$

Conversely, $G$ is trained to deceive $D$ by generating images that are not only misclassified but also indistinguishable from the actual data distribution.

$$\mathcal{L}_G = \mathbb{E}_{x \sim \mathcal{X}_0} \left[ \log D(G_{img}(x))_0 \right] + \mathbb{E}_{x \sim \mathcal{X}_1} \left[ \log D(G_{img}(x))_1 \right] + \mathbb{E}_{x \sim \mathcal{X}} \left[ \log D(G_{img}(x))_{\text{fake}} \right] \tag{2}$$

To improve $G$'s capability in producing counterfactual images, we incorporate an auxiliary classification loss $\mathcal{L}_{G_{cls}}$. This term essentially reflects the objective function of $D$, improving the ability of $G$ to determine the input's class label, which is essential for generating high-quality counterfactuals.

$$\mathcal{L}_{G_{cls}} = -\mathbb{E}_{x \sim \mathcal{X}_0} \left[ \log G_{cls}(x)_0 \right] - \mathbb{E}x \sim \mathcal{X}_1 \left[ \log G_{cls}(x)_1 \right] \tag{3}$$

Moreover, to ensure that $G$ produces minimally perturbed images, an L1 regularization term $\mathcal{L}_{G_s}$ is added. We selected the L1 norm for its effect of promoting sparsity in the perturbations, leading to more interpretable and localized changes.

$$\mathcal{L}_{G_s} = \mathbb{E}_{x \sim \mathcal{X}} [|x - G_{img}(x)|] \tag{4}$$

The overall objective function is thus a weighted sum of these individual loss components, where the weighting factors $\lambda_i$ are tunable hyperparameters.

$$\mathcal{L} = \mathcal{L}_D + \lambda_1 \mathcal{L}_G + \lambda_2 \mathcal{L}_{G_{cls}} + \lambda_3 \mathcal{L}_{G_s} \tag{5}$$

### 3.2.2 Cycle-consistent loss function

The adversarial nature of GANs often leads to training instability, particularly if the generator and discriminator evolve at different rates. Although, among other methods, Wasserstein GANs (Arjovsky et al., 2017) have proven effective at stabilizing GAN training via earth-mover's distance, they generally presuppose a univariate discriminator output. The discriminator outputs a 3D vector in our architecture, making the extension to a multi-variate Wasserstein loss non-trivial.

To counteract this limitation and enhance the gradient flow to $G$, we employ the cycle-consistent loss term, $\mathcal{L}_{G_c}$, similar to the method proposed by Charachon et al. (2022). However, while Charachon et al. relied on two generators for the domain translation, we employ a single model, $G$, to create nested counterfactuals (counterfactuals of counterfactuals) over $c$ cycles:

$$\mathcal{L}_{G_c} = \sum_{i=1}^{c} (-\lambda_c)^{i-1} \Bigg( \mathbb{E}_{x \sim \mathcal{X}_0} \left[ \log D(G_{img}^i(x))_0 \right] $$
$$+ \mathbb{E}_{x \sim \mathcal{X}_1} \left[ \log D(G_{img}^i(x))_1 \right] $$
$$+ \mathbb{E}_{x \sim \mathcal{X}} \left[ \log D(G_{img}^i(x))_{\text{fake}} \right] \Bigg) \tag{6}$$

Here, $c \in \mathbb{Z}_{\geq 1}$ represents the number of cycles, and $\lambda_c \in (0, 1]$ scales the influence of each half-cycle on the overall objective.

By doing so, the generator is exposed to a stronger gradient sourced from multiple cycles of counterfactuals, thereby resulting in more informed parameter updates. This cycle-consistent loss thus serves as an additional regularization term that bolsters both the training stability and the expressive capability of the generator.

### 3.2.3 Gradient update frequency

An additional strategy to maintain equilibrium between $G$ and $D$ involves carefully controlling the update frequency of each during the training process. By updating $D$ more frequently, we aim to ensure that $D$ is sufficiently accurate in distinguishing real from generated instances, thereby providing more meaningful gradient signals for $G$ to learn from.

# 4 EXPERIMENTS

## 4.1 DATASETS AND EVALUATION

**CASC IFW Database (2010)** (Li et al., 2009): This binary classification dataset contains over 5,800 apple images. It features an even split of healthy apples and those with Internal Feeding Worm (IFW) damage. Prior studies have conducted extensive architecture and hyperparameter optimizations (Ismail & Malik, 2022; Knott et al., 2023). Our experiments yield performances closely aligned with, though not identical to, these preceding works. To ensure rigorous evaluation, our adversarially trained models are compared to the re-implemented versions of the models rather than their reported metrics.

For this task, we selected the weight of the sparsity term $\lambda_3 = 0.1$ whereas all other terms are weighted by 1. Furthermore, $G$ received a gradient update for every second update of $D$.

**Concrete Crack Segmentation Dataset** (Özgenel, 2018): The Concrete Crack Segmentation Dataset features 458 high-resolution images of concrete structures, each accompanied by a binary map (B/W) that indicates the location of cracks for semantic segmentation. By cropping these images to the dimensions used by current CV models, it is possible to increase the dataset by approximately 50-fold. Our baseline model evaluations align with those reported in previous studies (Kim et al., 2021; Ali et al., 2022), validating our implementation approach.

Here, we opted for a higher value of $\lambda_3 = 2$ to ensure consistency between input and counterfactual images. The other terms are again weighted by 1. $G$ and $D$ were updated with the same frequency.

## 4.2 DATA PREPROCESSING

We use a consistent data augmentation pipeline for both datasets, including random cropping and resizing to 224x224 pixels. We also apply slight brightness, hue, and blur adjustments, along with randomized flipping and rotations, to enhance diversity in the dataset and bridge the gap between real and generated data distributions. None of the two datasets provide predefined training, validation, and test splits. However, according to the previous works mentioned above, we randomly split them into subsets of size 70%, 10%, and 20%, whereby the splitting is performed prior to image cropping to avoid data leakage.

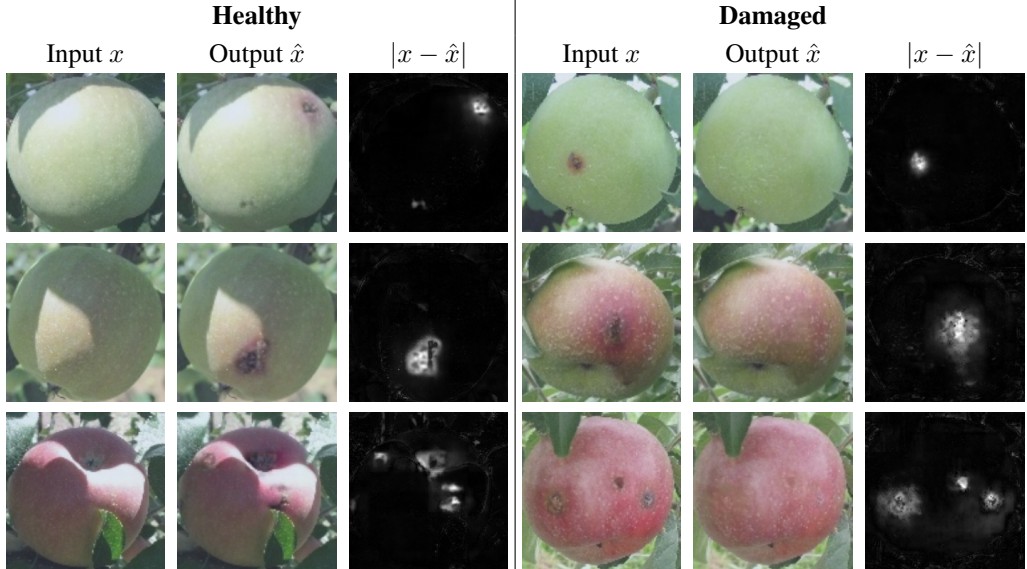

Figure 2: Examples of counterfactual images on the CASC IFW test set generated by our Swin UNet $G$ trained in conjunction with a Hybrid $D$. The input $x$ is fed to the generator $G$, which produces the counterfactual $\hat{x}$ with its $G_{\text{img}}$ branch. The absolute difference between $x$ and $\hat{x}$ highlights the salient regions in the image.

## 4.3 COUNTERFACTUAL IMAGE QUALITY ASSESSMENT

Figure 2 shows counterfactual examples of the Swin UNet-based $G$ when trained alongside a Hybrid $D$. The model demonstrates remarkable efficacy in synthesizing counterfactual images by either introducing or eradicating apple damage with high fidelity.

Besides qualitative visual evaluation, we also perform a quantitative analysis by calculating the Fréchet inception distance (FID) between generated images and real images from the dataset. Table 1 shows the influence of different architectural combinations for both $G$ and $D$ on the quality of produced counterfactuals. Importantly, the employment of cycle consistency loss positively impacts convergence, enhancing the model's robustness against mode collapse. Furthermore, the comparative analysis clearly demonstrates the superiority of the Swin UNet backbone for $G$ over its CNN UNet counterpart.

Table 1: Comparison of Fréchet inception distance (FID) for different Generator-Discriminator combinations with and without cycle-consistency loss on the CASC IFW Database. Missing FID entries indicate combinations of $G$ and $D$ did not converge due to a mode collapse.

| $G$  ⟍  $D$ | Cycles | **ResNet3** | **ResNet18** | **ResNet50** | **Swin Small** | **Hybrid** |
|---|---|---|---|---|---|---|
| **CNN UNet** | 0 | 0.086 | 0.205 | - | - | - |
|  | 1 | 0.064 | 0.049 | 0.168 | - | - |
| **Swin UNet** | 0 | 0.072 | 0.162 | 0.139 | - | 0.021 |
|  | 1 | 0.073 | 0.043 | 0.114 | 0.021 | **0.016** |

All subsequently reported results for $G$ and $D$ were obtained with models that included the cycle consistency term in their loss function.

## 4.4 CLASSIFICATION PERFORMANCE

We investigate the classification performance of both $G$ and $D$ under various architectural backbones, including ResNet variants and Swin Transformers. These models are compared against their non-adversarially trained equivalents on a range of classification metrics.

Table 2: Classification Metrics on CASC IFW test split. Models $G$ and $D$ employ our counterfactual pipeline; equivalent models conventionally trained for classification.

| Model | Accuracy | F1-Score | Precision | Recall |
|---|---|---|---|---|
| ResNet18 | 0.932 | 0.946 | 0.946 | 0.946 |
| ResNet50 | 0.937 | 0.949 | 0.955 | 0.944 |
| Swin Small | 0.978 | 0.983 | 0.982 | 0.983 |
| Hybrid | **0.980** | **0.984** | 0.984 | 0.985 |
| CNN UNet | 0.924 | 0.942 | 0.903 | 0.985 |
| Swin UNet | **0.980** | 0.984 | 0.981 | 0.986 |
| ResNet18 $D$ | 0.836 | 0.853 | 0.967 | 0.763 |
| ResNet50 $D$ | 0.866 | 0.885 | 0.954 | 0.826 |
| Swin Small $D$ | 0.952 | 0.963 | 0.937 | **0.990** |
| Hybrid $D$ | 0.979 | 0.983 | 0.979 | 0.987 |
| CNN UNet $G$ | 0.919 | 0.936 | 0.923 | 0.950 |
| Swin UNet $G$ | 0.979 | 0.983 | **0.987** | 0.980 |

The performance summary, presented in Table 2 reveals that the adversarial training routine does not lead to a significant drop in accuracy and that the models employing a Swin Transformer as backbone yield a better performance over ResNet-based models.

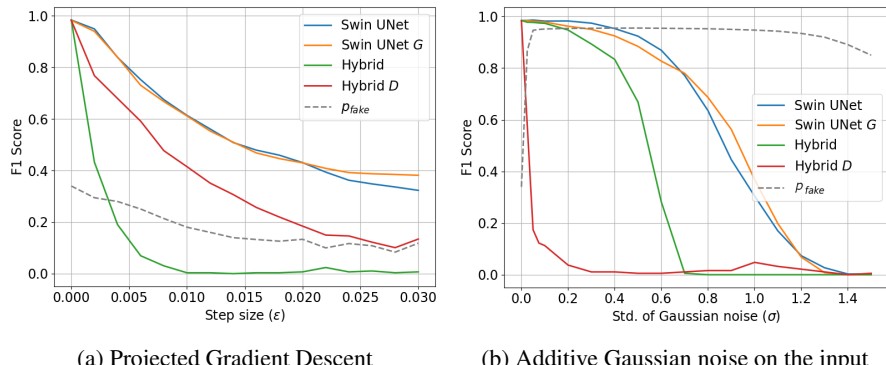

(a) Projected Gradient Descent          (b) Additive Gaussian noise on the input

Figure 3: Effects of two perturbation methods on the F1 scores of our best-performing model and its equivalent components trained on the typical classification task. The predictions of $D$ were obtained by taking the $\mathrm{argmax}$ between $p_0$ and $p_1$. The third value, $p_{\mathrm{fake}}$, is also depicted with a dashed grey line.

### 4.5 ROBUSTNESS AGAINST ADVERSARIAL ATTACKS

Figure 3 shows the effects of adding perturbations to the input images by plotting the strength of the attack against the F1-score: step size for PGD over 10 iterations with 0.1 maximum perturbation at $L_\infty$ norm using the Adversarial Robustness Toolbox Nicolae et al. (2018) (left), and standard deviation for added Gaussian noise (right) . Figure 3a shows that $D$ with a "Hybrid" architecture is more robust compared to its non-adversarial counterpart. Since we performed targeted PGD attacks, the "fakeness" value $p_{\mathrm{fake}}$ does not yield insight into the attack's strength. On the other hand, it is highly effective in detecting noise, as shown in Figure 3b. $G$ maintains comparable robustness to both PGD and noise injection without a significant difference in F1-score relative to a non-adversarial Swin UNet.

Regarding the observed decline in F1-score to zero as perturbations intensify, this might initially appear counterintuitive, given that two random vectors would yield an expected F1-score of 0.5. However, it's important to note that these models are specifically fine-tuned to identify defects. At higher noise levels, defects become statistically indistinguishable from undamaged instances, causing the model to label all samples as undamaged, resulting in zero true positives and thus an F1-score approaching zero.

We conducted a comprehensive evaluation to determine the effectiveness of the "fakeness" value $p_{\mathrm{fake}}$ in measuring a model's prediction confidence. Our methodology involved calculating the negative log-likelihood loss per sample and comparing it to the Pearson correlation coefficient with $p_{\mathrm{fake}}$. After analyzing our Swin UNet $G$ and Hybrid $D$ models, we found that the class predictions had a coefficient of 0.081 and 0.100, respectively. These results indicate that the loss is positively correlated with $p_{\mathrm{fake}}$, which can therefore serve as a dependable measure of model uncertainty at inference time. More details on our analysis and calculations are provided in Appendix A.3.

### 4.6 SALIENCY MAP QUALITY ASSESSMENT

We evaluate if the absolute difference $|G_{\mathrm{img}}(x) - x|$ between the input image $x$ and its counterfactual $G_{\mathrm{img}}(x)$ are effective at highlighting salient regions in the image on the Concrete Crack Segmentation Dataset. Figure 4 shows that both CNN and Swin UNet $G$ models trained with our adversarial framework produce saliency masks that are more accurate and predictive compared to GradCAM. In fact, the SwinUNet $G$ generates highly localized and contrast-rich saliency masks, closely resembling those produced by segmentation models trained on pixel-level annotations. The similar quality between masks produced by models trained with pixel-level labels and our adversarial models can be quantified when computing the IoU values between the masks and the ground-truth segmentation masks. Figure 5 shows that our SwinUNet $G$ reaches IoU scores merely 12% lower than the best-performing segmentation models despite never having seen pixel-level annotations. On the other hand, the other attribution method, GradCAM, reaches IoU scores well below ours.

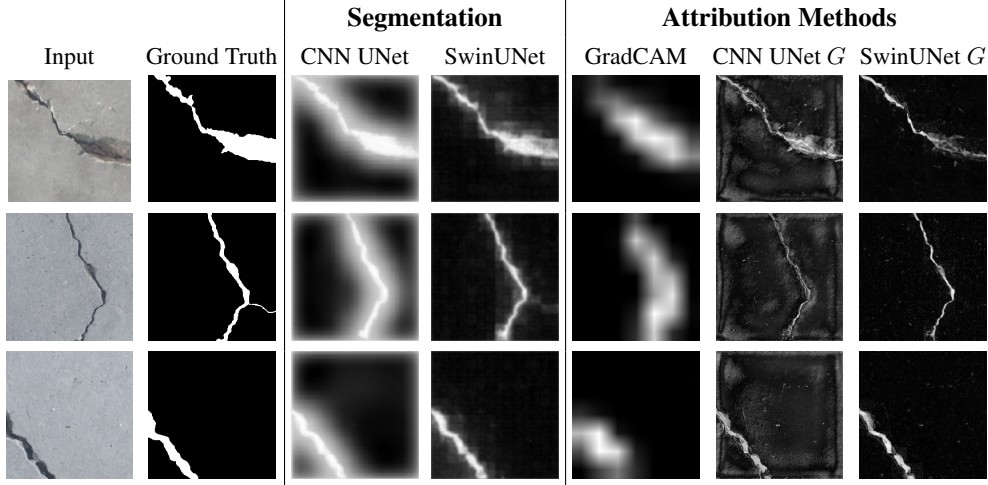

Figure 4: Segmentation mask comparison for concrete cracks. Models trained on pixel-level annotations are shown in the two center columns, while those trained on classification labels are displayed in the three right-most columns. CNN UNet $G$ and Swin UNet $G$ are our adversarial models.

## 5 DISCUSSION

The proposed framework shows great potential in producing high-quality saliency maps despite relying only on classification labels. Annotating a dataset with class labels instead of segmentation masks requires significantly less effort. This allows the assembly of larger and more diverse datasets, potentially further narrowing the gap between segmentation and attribution methods.

However, this implementation is constrained to binary classification tasks. While multi-class adaptation techniques exist (Charachon et al., 2022), we argue that high-quality explanations are best generated through comparisons of each class against the background. Therefore, we would address a multi-class problem by breaking it into several binary classification objectives. Additionally, the method requires a balanced dataset for stable training of both $G$ and $D$, which can be problematic in anomaly and defect detection contexts where datasets are notoriously imbalanced.

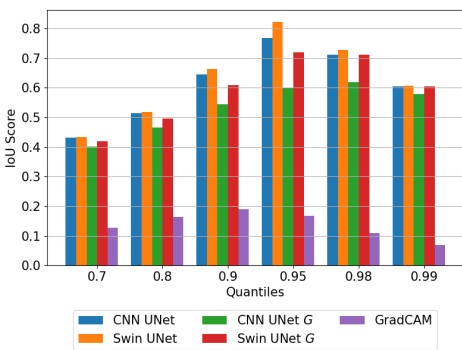

Figure 5: IoU Scores for concrete crack segmentation using pixel-level annotations in CNN- and Swin UNet, and class labels for CNN UNet $G$, Swin UNet $G$, and GradCAM. Scores are plotted across varying mask binarization thresholds.

## 6 CONCLUSION

Our research presents a unified framework that addresses both interpretability and robustness in neural image classifiers by leveraging image-to-image translation GANs to generate counterfactual and adversarial examples. The framework integrates the classifier and discriminator into a single model capable of both class attribution and identifying generated images as "fake". Our evaluations demonstrate the method's efficacy in two distinct domains. Firstly, the framework shows high classification accuracy and significant resilience against PGD attacks. Additionally, we highlight the role of the discriminator's "fakeness" score as a novel uncertainty measure for the classifier's predictions. Finally, our generated explainability masks achieve competitive IoU scores compared to traditional segmentation models, despite being trained solely on classification labels.

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

# A  APPENDIX

## A.1  GENERATIVE UNET ARCHITECTURE

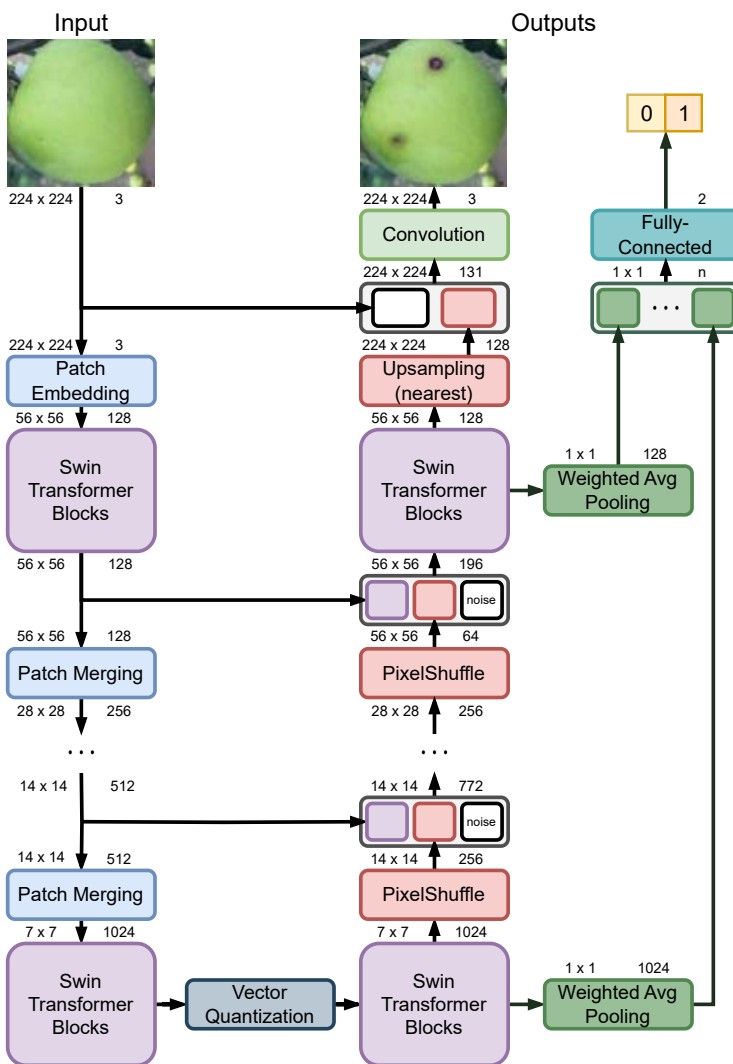

Figure 6: Our adapted SwinUNet architecture. Modifications compared to the architecture proposed by Cao et al. (2022) involve the use of a vector quantization layer, concatenated noise at every upsampling layer, replacing PixelShuffle with nearest neighbor upsampling the last block and a class probability prediction branch using weighted average pooling and a fully-connected network.

We experiment with two different architectures for the generator: the classical CNN-based UNet architecture and a modified version, Swin UNet (Cao et al., 2022; Fan et al., 2022), where the convolutions are replaced with Swin Transformer blocks.

We incorporate stochasticity into the model in both architectures by introducing noise at each upsampling layer. Specifically, a noise vector $z$ is sampled from a standard normal distribution, i.e., $z \sim \mathcal{N}(0, I)$, where the dimensionality of $z$ is $d$, a model hyperparameter. This vector is subsequently linearly projected into a higher-dimensional space of dimensions $\mathbb{R}^{H_i \times W_i \times c}$, where $H_i$ and $W_i$ denote the height and width of the feature map at a given layer $i$, and $c$ represents the number of channels introduced to the feature map, another model hyperparameter. The noise is then concatenated to the feature maps at each upsampling layer.

Let $u_i \in \mathbb{R}^{H_i \times W_i \times C_i}$ represent the $i$-th upsampling layer of the U-Net architecture. The noise is concatenated to the feature map as follows:

$$u_i = W_{1i}\text{Concat}(u_i, W_{0i}z) \tag{7}$$

where $W_{0i}$ is a weight matrix of dimensions $c \times H_i \times W_i \times d$, and $W_{1i}$ a weight matrix of dimensions $C_i \times H_i \times W_i \times (C_i + c)$, thus projecting the feature map back to its original dimensions.

This modification is designed to induce variation in the model's output. Past research has indicated that when noise is incorporated solely at the bottleneck of the generator, the model often neglects this noise during the learning process Isola et al. (2017). By strategically injecting small amounts of noise at each upsampling stage, the generator is compelled to accommodate this noise more attentively, resulting in a model capable of generating a more robust and diverse array of images.

$G$ is equipped with an auxiliary classification objective, which is implemented by adding a secondary branch after weighted average pooling from each upsampling block. This secondary branch consists of a fully connected network that predicts the class label of the input image. The pooling operation is realized through a two-branch scoring and weighting mechanism (Yang et al., 2022).

For the Swin UNet, we made further modifications such as the use of a Swin Transformer pretrained on Imagenet (Russakovsky et al., 2015) as the encoding part of the UNet. Additionally, we employed a Vector Quantization layer at the bottleneck, inspired by current state-of-the-art image-to-image translation models (Rombach et al., 2022). The adapted Swin UNet architecture is illustrated in Figure 6.

## A.2 COUNTERFACTUAL IMAGE QUALITY ASSESSMENT

Table 3: Comparison of Fréchet Inception Distance (FID) across various Generator-Discriminator architectures on the CASC IFW Database, under conditions with and without cycle-consistency loss. "Set" specifies which images were included in the calculation: "full" means all images and counterfactuals, "und." includes all undamaged images and counterfactuals from damaged images, "dam." are all damaged images and counterfactuals from undamaged images. Missing FID values indicate non-convergence due to mode collapse.

| $G$ $\diagdown$ $D$ | Cycles | Set | ResNet3 | ResNet18 | ResNet50 | Swin Small | Hybrid |
|---|---|---|---|---|---|---|---|
| **CNN UNet** | 0 | full | 0.086 | 0.205 | - | - | - |
| | | und. | 0.177 | 0.302 | - | - | - |
| | | dam. | 0.176 | 0.293 | - | - | - |
| | 1 | full | 0.064 | 0.049 | 0.168 | - | - |
| | | und. | 0.171 | 0.182 | 0.272 | - | - |
| | | dam. | 0.169 | 0.165 | 0.269 | - | - |
| **Swin UNet** | 0 | full | 0.072 | 0.162 | 0.139 | - | 0.021 |
| | | und. | 0.192 | 0.211 | 0.251 | - | 0.178 |
| | | dam. | 0.178 | 0.254 | 0.285 | - | 0.159 |
| | 1 | full | 0.073 | 0.043 | 0.114 | 0.021 | **0.016** |
| | | und. | 0.189 | 0.157 | 0.174 | 0.163 | **0.139** |
| | | dam. | 0.168 | 0.164 | 0.321 | 0.183 | **0.132** |

Table 3 contains all FID scores computed over three different subsets of the dataset: in "full", all real images were compared against all counterfactual images, in "und.", undamaged images were compared against counterfactuals that originated from damaged images and should now not contain damages anymore, and "dam." contains all damaged images and counterfactuals from undamaged images which should now contain damage. Note that the scores on "dam." and "und." are expectedly worse because they do not contain both the real image and its counterfactual, whereas "full" does.

A.3 FAKENESS AS UNCERTAINTY ESTIMATION

The objective is to determine the degree to which $p_{\text{fake}}$ correlates with the model's prediction error as quantified by the negative log-likelihood loss.

$$l_{\text{nll}}(p_1^{(i)}, y^{(i)}) = - \left[ y^{(i)} \log(p_1) + (1 - y_i) \log(1 - p_1) \right] \tag{8}$$

To calculate the cross-correlation coefficient $r$ we employ the Pearson correlation coefficient formula:

$$r = \frac{\sum_{i=1}^{N} (l_{\text{nll}}^{(i)} - \bar{\mathcal{L}}_{\text{nll}})(p_{\text{fake}}^{(i)} - \bar{p}_{\text{fake}})}{\sqrt{\left( \sum_{i=1}^{N} (l_{\text{nll}}^{(i)} - \bar{\mathcal{L}}_{\text{nll}})^2 \right) \left( \sum_{i=1}^{N} (p_{\text{fake}}^{(i)} - \bar{p}_{\text{fake}})^2 \right)}} \tag{9}$$

where $N$ represents the total number of samples, $\bar{\mathcal{L}}_{\text{nll}}$ and $\bar{p}_{\text{fake}}$ are the average negative log-likelihood loss and average "fakeness" value across all samples, respectively.

A positive value of the cross-correlation coefficient $r$ would indicate that $p_{\text{fake}}$ is a reliable indicator of the model's prediction confidence, while a value close to 0 would suggest otherwise.

Table 4: Pearson correlation coefficient calculated between the negative log-likelihood loss on predictions from $G$ and $D$ against the uncertainty measure $p_{\text{fake}}$ produced by $D$ on the CASC IFW Database for all combinations of $G$ and $D$ backbone architectures.

| $G$ \ $D$ | | ResNet3 | ResNet18 | ResNet50 | Swin Small | Hybrid |
|---|---|---|---|---|---|---|
| **CNN UNet** | $G_{\text{cls}}(x)$ | -0.014 | -0.011 | 0.007 | - | 0.062 |
| | $D(x)$ | -0.003 | 0.071 | 0.197 | - | 0.118 |
| **Swin UNet** | $G_{\text{cls}}(x)$ | 0.011 | 0.012 | 0.003 | 0.067 | 0.073 |
| | $D(x)$ | 0.169 | 0.060 | 0.130 | 0.064 | 0.100 |

When examining the correlation values presented in Table 4, it becomes evident that most of them exhibit positive correlations, except for a few weaker models. This observation implies that the "fakeness" value obtained from $D$ can be effectively employed during the inference process to ascertain the model's prediction confidence level.

## A.4  Counterfactual images for concrete cracks

| Input $x$ | Output $\hat{x}$ | $|x - \hat{x}|$ | Input $x$ | Output $\hat{x}$ | $|x - \hat{x}|$ |

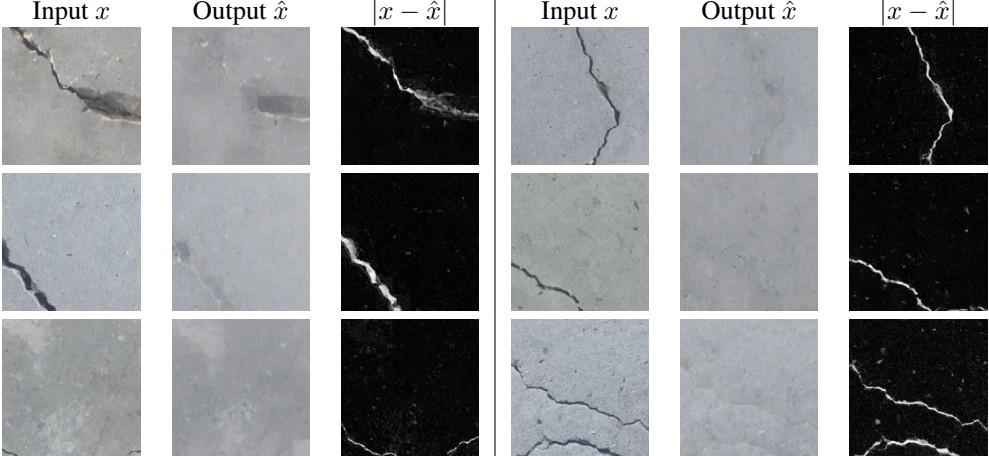

Figure 7: Examples of counterfactual image generation on the Concrete Crack Dataset. The input $x$ is fed to the generator $G$, which produces the counterfactual $\hat{x}$ with its $G_{\text{img}}$ branch. The absolute difference between $x$ and $\hat{x}$ highlights the salient regions in the image.

Table 5: Comparison of Segmentation Scores on the Crack dataset test split. Reported values are the maximum over the quantiles in Fig. 5.

| Model | Accuracy | F1-Score | IoU |
|---|---|---|---|
| GradCAM | 0.954 | 0.341 | 0.206 |
| UNet | 0.975 | 0.976 | 0.780 |
| Swin UNet | 0.981 | 0.982 | 0.825 |
| UNet ($G*$) | 0.971 | 0.964 | 0.622 |
| Swin UNet ($G*$) | 0.976 | 0.974 | 0.720 |

*$G$ are generators and $D$ discriminators trained in our adversarial setting

