# OpenReview forum: "Counterfactual Image Generation for adversarially robust and interpretable Classifiers"
_ICLR.cc/2024/Conference — Submitted to ICLR 2024_

### Official Review · Reviewer_9wk6 · 2023-10-23

**Soundness:** 1 poor
**Presentation:** 2 fair
**Contribution:** 3 good
**Rating:** 3
**Confidence:** 4

**Summary:**

This paper presents a unified framework for learning interpretable and adversarially robust binary classifiers. The proposed approach combines the training of a GAN with counterfactual images. The paper then presents results of binary classification performance using either the Generator or Discriminator, and binary segmentation mask using the Generator. The Generator and Discriminator have similar classification performance than unmodified baselines, but are more robust to adversarial examples. Finally, binary masks obtained from the difference between counterfactual and original images are sharper than GradCAM.

**Strengths:**

- **Noteworthy Contribution**: The paper introduces a novel framework that has the potential to address the challenges of achieving robust and interpretable classifiers. This contribution is particularly relevant in the context of existing classifiers that often struggle with the trade-off between robustness and interpretability.

- **Varied Evaluation**: The paper's assesses the proposed method using different model architectures, including convolutional networks and transformer models. This varied evaluation demonstrates the adaptability of the approach across different scenarios and model types, highlighting its potential for broader applicability.

**Weaknesses:**

- **Limited to Binary Tasks**: A major limitation of the paper is that it only addresses binary classification tasks. It would be interesting to expand its applicability to multiclass problems to demonstrate broader utility, as mentioned in the discussion section.

- **Single Seed Experiments**: The experiments in the paper are limited to training on a single seed, making it difficult to assess the significance of performance differences and the true impact of the proposed cycle consistency loss on convergence. Multiple seed experiments would provide a more robust evaluation.

- **Experiment Clarity**: The presentation of experiments can be confusing and should be more detailed. For instance, the "Hybrid D" model is never introduced in the paper. The explanation of the computation of performance when using D is also presented *after* showing results. The description of Table 2 is also unclear, making it challenging for readers to understand the methodology and the comparison.

- **Misleading Introduction**: The paper introduces the approach as "combining classifier and discriminator in a single model" (in the abstract), which is incorrect since the generator and discriminator are fundamentally different.

- **Lack of Comparative Analysis**: The paper lacks a comparison with other counterfactual approaches, which could provide insights into the quality of the counterfactuals produced and help position the proposed method within the broader context of counterfactual research.

**Questions:**

- It could be interesting to have a rule of thumb in which model to use, G or D ? Both seems to be strong for classification, but do they have their own advantages ?
- Can we do more than 1 cycle in the "cycle consistency loss" ? The loss is described with $c \geq 1$, but experiments only show $c=1$ or $c=0$.
- I'm not sure about the conclusion of section 4.5 on the robustness of the classifiers. From Figure 3, we can see that the models trained with the proposed approaches both have *lower* F1 scores than the baselines when increasing the perturbation size. I assumed that the labels of the approaches are inverted in the plot. Can the authors clarify this ? Otherwise, the conclusion on robustness would be completely different.

---

> ### Author Response · Authors · 2023-11-22
>
> Thank you for your review. We appreciate the time and effort you have put into reading our work and communicating your assessment.
>
> We would like to address the points mentioned in "Weaknesses":
> 1. We acknowledge that being limited to binary classification may seem like a limitation. However, this setting covers a wide range of problems in medical health diagnosis, structural health monitoring, and other tasks involving anomaly detection. Furthermore, we argue that visual explanations of class predictions are most informative when compared against a “background” class, as, for example, directly comparing an image of a table against an image of a chair may not yield the best visual explanation of what a chair is. As such, our framework would be applicable to multi-class problems when formulating them as several, separate binary classification tasks where each class is either present or absent from the image in question.
>
> 2. We acknowledge the importance of this point. In our experiments, although there was some variability in the perceived quality of counterfactual image generations across different runs, the classification performance remained mostly consistent. Our primary focus was on demonstrating the utility of counterfactual image generation for image classifiers, rather than on classification performance per se. Hence, we prioritized showcasing other metrics in the limited space available. Additionally, given the short timeframe for rebuttal and the substantial training time required (12 to 24 hours for larger models), systematically re-training all models to collect comprehensive multi-seed data was not feasible.
>
> 3. The Hybrid discriminator is introduced in Section 3.1.2. We added a sentence explicitly stating the name that will be used for the hybrid architecture in the revised version.
>
> 4. We indeed combined the classifier and discriminator into a single model, as it has outputs for estimating the “fakeness” of an image as well as one for the class prediction. The confusion may come from the fact that we added the auxiliary task of classifying the images to the generator as well. In our framework, both the generator and discriminator make predictions for the class.
>
> 5. We acknowledge the absence of a comparative analysis with other counterfactual approaches in our paper. Our challenge was the scarcity of comparable counterfactual image generation methods that reported performance metrics aligning with our focus, such as saliency map evaluation against segmentation labels. Additionally, re-implementing existing methods with sufficient fidelity for an accurate performance comparison proved to be complex. As such, we focused on demonstrating the efficacy of our method within the constraints of available comparative data and reproducibility standards.
>
> Concerning the questions:
> 1. They do indeed both have good classification performances. If efficiency is important, then the discriminator is the better choice due to its smaller architecture size. If the saliency maps are always necessary, then we would suggest using only the generator. However, the one would obtain the best performing model by combining the generator's and discriminator's prediction as kind of ensemble.
> 2. There is no limit to the number of cycles that could be used until hardware limitations (memory) are reached. We experimented with values higher than 1, but the minor improvements did not justify the longer training times.
> 3. You are of course correct. This mishap must have occurred during last-minute updates to the Figure's legend in an effort to make all legends consistent. We corrected the mistake and uploaded the new Figure.

---

> > ### Comment · Reviewer_9wk6 · 2023-11-22
> >
> > Thank you for the answer and the additional work.
> > 1. I agree that binary classification is used in a variety of applications. However, so is multi-class classification, and multi-class problems are usually more complex than binary ones. I understand how the proposed approach could be applied in the multi-class setting, but it remains to be proven that it remains effective. If so, the significance of the work would be greatly improved.
> > 2. I understand that training on more seeds in the time frame of the rebuttal is complicated. However, the concerns mentioned in the initial review remain.
> > 3. Thank you for the clarifications on this part.
> > 4. Indeed, there is confusion because the generator also classifies images, making it a second classifier in the approach. However, there is also initially no other classifier in GANs than the discriminator, so mentioning "the classifier and discriminator" is unclear and leads to confusions. In my opinion, a better wording would be something like: "This is achieved by extending the task of the discriminator to attributing real images to their respective classes, while flagging generated images as ”fake”. ".
> > 5. I agree with the authors that it is difficult to compare to other counterfactual image generations methods. However, it is currently difficult to assess the significance of the results without other strong baselines to compare to. The results cannot be properly evaluated only with few qualitative examples and numerical values in a vacuum. Furthermore, as discussed by the authors in their related work section, there are more recent attribution methods than GradCAM to compare to.
> >
> > I find the proposed idea, the approach, and the objectives interesting, but the quality of the empirical comparison needs to be improved for acceptance. For this reason, I stand by my rating.

---

### Official Review · Reviewer_jyYD · 2023-10-28

**Soundness:** 1 poor
**Presentation:** 3 good
**Contribution:** 2 fair
**Rating:** 3
**Confidence:** 4

**Summary:**

This paper proposes an explanation framework using image-to-image GAN, whose discriminator obtains adversarial robustness during training. The generator in the framework learns the visual transformation between binary labels (e.g., the healthy apple and the damaged apple). Authors claim that the absolute pixel difference from the transformation reflects model explainability and the discrimination process helps the target model obtain adversarial robustness.

**Strengths:**

1. The paper is well-presented with a storyline that demonstrates the proposed framework.
2. The paper evaluates multiple types of model structures, including CNN and Transformer. The coverage of experiments on target model structure is comprehensive.
3. The topic of trustworthiness analysis with generative models is increasingly important.

**Weaknesses:**

1. In Section 4.5, the authors mention "...adding perturbations to the input images by plotting the strength of the attack (**step size** for PGD over 10 iterations..." This setup is questionable. For a regular PGD attack, the **perturbation bound** is a more direct reflection of the attack strength. However, in this section, the authors plot the F1 score w.r.t. the attack step size (Figure 3a), which is less informative for showing attack strength. Also, the authors should state clearly in the PGD experiment setup what value the perturbation bound takes.

2. The framework is a modification (changing the discriminator objective) derived from the training process of an image-to-image GAN. The claim of simply using absolute pixel differences of image translation as counterfactual explainability is not grounded. It only visualizes the changing semantics and is not always sufficient as a counterfactual explanation to the target classifier. Recent years have witnessed more effective and solid approaches to generating semantic counterfactuals/adversaries [1,2,3,4,5,6]. The user can perform adversarial training with these methods to fine-tune and improve the target classifier. The paper should compare with more baselines from this line of research.

3. There are other generative paradigms like diffusion models and VAEs, which have shown the capability to perform image-to-image translation. It can also be feasible that we jointly train an image-to-image diffusion model with the target classifier. The paper should show sufficient validity in adopting the GAN paradigms (e.g., convergence, efficiency) in the experiments.

[1] (ICCV 2019) Semantic Adversarial Attacks: Parametric Transformations That Fool Deep Classifiers.

[2] (ECCV 2020) SemanticAdv: Generating Adversarial Examples via Attribute-conditional Image Editing.

[3] (ICCV 2021) Explaining in Style: Training a GAN to explain a classifier in StyleSpace.

[4] (CVPR 2023) Zero-Shot Model Diagnosis.

[5] (CVPR 2023) Adversarial Counterfactual Visual Explanations.

[6] LANCE: Stress-testing Visual Models by Generating Language-guided Counterfactual Images.

**Questions:**

1. What is the possible performance of using other GAN paradigms (e.g., StyleGAN variants) compared to the proposed approach (CycleGAN variants) on this task?
2. Please address my concerns stated in the weakness section. I would revise the rating based on further responses/rebuttals from the authors.

---

> ### Author Response · Authors · 2023-11-22
>
> Thank you for your review. We appreciate the time and effort you have put into reading our work and communicating your assessment.
>
> We would like to address the points mentioned in "Weaknesses":
>
> 1. There was indeed important information missing. We added the values used for the perturbation bound, norm, and codebase used for the attack. We selected a rather small L_inf bound of 0.1 because we are interested in small, imperceptible modifications of the image.
>
> 2. The generator’s objective is to create an image belonging to the opposite class compared to the original image and is as such trained to produce counterfactuals. The information about the semantics of the two classes is provided by the gradients flowing through the discriminator. Indeed, this information may be imperfect when the discriminator focuses on non-class-relevant features, like radiologist annotations in medical images. In such scenarios, our generator might include these annotations in healthy samples, creating images that are not necessarily counterfactuals in real-world terms, but still counterfactuals relative to the dataset and the discriminator’s predictions. The evaluation on segmentation masks yields a quantitative metric for the produced counterfactual samples and can reveal such “Clever Hans” situations.
>
> 3. Indeed, other models could be excellent at generating counterfactual images as well. However, the counterfactual image generation per se is only a subtask in our study, which aims at merging two lines of work: one focusing on using GANs for creating counterfactual images and one using GANs for creating adversarially robust classifiers. As such, our models are not comparable to diffusion models or VAEs.
>
> Concerning the question on other GAN paradigms: our method could be used in combination with other GAN variants that aim at improving training stability and output quality, such as StyleGAN. We adopted the CycleGAN approach because it is an intuitive addition and has already been proposed for counterfactual image generation: Charachon et al., “Leveraging conditional generative models in a general explanation framework of classifier decisions”.

---

> > ### Comment · Reviewer_jyYD · 2023-11-22
> >
> > This submission needs further revising to reach the bar of acceptance. Given the response that only partially addresses my questions and concerns, I will rate this paper a rejection.

---

### Official Review · Reviewer_ujhc · 2023-10-31

**Soundness:** 1 poor
**Presentation:** 1 poor
**Contribution:** 1 poor
**Rating:** 1
**Confidence:** 4

**Summary:**

Even though both adversarial attack and interpretability are important in classification, existing methods exclusively address either one of them. The authors propose a framework for training a classifier that is interpretable and robust against adversarial attack. The framework is designed to be end-to-end, and its performance is shown in both quantitative and qualitative experiments.

**Strengths:**

Interesting idea merging adversarial attack and interpretability — proposing a classifier robust against adversarial attack and interpretable.

**Weaknesses:**

1. Writing is unclear.
    1. (page 2) “we introduce a unified framework that merges the generation of adversarial samples for enhanced robustness with counterfactual sample generation for improved interpretability.“ => “we introduce a unified framework that merges the generation of adversarial samples for enhanced robustness and improved interpretability with counterfactual sample generation.“
    2. (page 2) “to minimally alter the images such that they are classified into the opposing class” => What is the opposing class?
    3. (page 1) “We argue that by fixing the classifier’s parameters, current attribution methods using GANs forfeit the opportunity to train a more robust classifier simultaneously, even though it has been previously observed that adversarial attacks could also be employed as tools for interpreting the model’s decision-making process“ => What is the “even though" sentence for?
    4. (page 2, page 4) “This methodology has the benefits of (i) creating adversarial examples that augment the dataset, making the classification more robust against subtle perturbations“ => augmenting dataset can be done after training Generative Models and making the classifier more robust can be done during the training (under the method in this paper).
    5. (Figure 1, page 4) “Conversely, G must deceive D by producing realistic samples attributed to the opposite class by D.” => What does this mean?
    6. (page 4) “${\hat{x}=G(x)}$ that is misclassified by D: not only should D be unable to detect that the counterfactual was generated by G, it should also attribute it to the wrong class.“ => This is not understandable and seemingly incorrect.
    7. (page 4) in Section 3.1.1, the authors mention that ${\hat{x}=G(x)}$  and in Section 3.2, ${G(x)=(\hat{x},\hat{y})}$.

2. The terminology “Counterfactual” is used in an unreasonable way; how is damage/no-damage related to factual/counterfactual?
3. Novelty is limited;
    1. it seems like the proposed method is a simple variant of ACGAN [1,2].
    2. only binary classification is discussed.


[1] StarGAN, CVPR'18
[2] Conditional image synthesis with auxiliary classifier gans, ICML'17

**Questions:**

No questions.

I would recommend
1. make sure that the authors understand the proposed method.
2. define the task clearly (ideally with the task used in the experiment).
3. writing straightforward rather than beating around the bush.

---

> ### Author Response · Authors · 2023-11-22
>
> Thank you for your review. We appreciate the time and effort you have put into reading our work and communicating your assessment.
>
> We would like to address the points mentioned in "Weaknesses":
>
> 1.2. Our method is designed for binary image classification tasks, which involve two distinct classes. In this context, the 'opposing class' refers to the alternative category that an image does not currently belong to. The aim is to achieve minimal alterations to the original image to transition it convincingly into this opposite class.
>
> 1.3. Our critique of current explainability methods using GANs is that they miss an opportunity for dual improvement. By fixing the classifier's weights during the adversarial generation phase, these methods manage to highlight a classifier’s proneness to small perturbations but do not offer a solution for how to overcome this limitation. Our proposed approach addresses this gap. \
> We acknowledge that this paragraph lacked some clarity and revised it in the new version.
>
> 1.4. Unlike pre-trained generative models, our methodology creates targeted augmentations in an online fashion during training. The generator in our framework is designed to produce minimally altered images that are misclassified by the discriminator in terms of both realism and class attribution. As the generator and discriminator are trained together in a min-max game, these augmented samples progressively improve. This leads to increasingly refined counterfactual samples.
>
> 1.5. This is probably related to a previous question: in this work, we focused on binary image classification. The second important aspect of our work is the fact that we combine the classifier and discriminator in a single network. It must predict the class of the image and whether the image was generated or not. The generator, on the other hand, must create images that the discriminator (i) misclassifies (attributes to the wrong class) and (ii) fails to detect that it was generated (outputs a high probability that the image belongs to the real distribution although it was generated).
>
> 1.6. Again, this probably becomes more clear when narrowing down the problem to a binary classification task.
>
> 2.1. We use the terminology of a of research leveraging counterfactual images as explainability for neural image classifiers:\
> CounteRGAN: Generating Realistic Counterfactuals with Residual Generative Adversarial Nets, Daniel Nemirovsky et al.\
> GANterfactual—Counterfactual Explanations for Medical Non-experts Using Generative Adversarial Learning, Silvan Mertes et al. \
> EXPLAINING IMAGE CLASSIFIERS BY COUNTERFACTUAL GENERATION, Chang et al.
> We included additional citations of related work in the area of counterfactual image generation as additional information for the readers.
>
> 3.1. ACGANs indeed share similarities with our method in integrating the discriminator and classifier. However, our approach differs significantly, primarily in our generator's design, which is unconditioned, as opposed to the class-conditioned generator in ACGANs. This distinction is crucial for generating explanations in our framework, where we compare the original image directly with its counterfactual generated by G. To achieve something similar with ACGANs, one would have to obtain the latent representation z of the original image, generate a new image conditioned on class 0, and another on class 1, and then compare these two generated images. In this scenario, one would be comparing two generated images, where important features from the original image may have been lost or modified. ACGANs were not designed to solve the same problem that we are targeting.
>
> 3.2. We acknowledge that being limited to binary classification may seem like a limitation. However, this setting covers a large range of problems in medical health diagnosis, structural health monitoring, and other tasks involving anomaly detection. Furthermore, we argue that visual explanations of class predictions are most informative when compared against a “background” class, as, for example, directly comparing an image of a table against an image of a chair may not yield the best visual explanation of what a chair is. As such, our framework would be applicable to multi-class problems when formulating them as several, separate binary classification tasks where each class is either present or absent from the image in question.

---

> > ### Comment · Reviewer_ujhc · 2023-11-22
> >
> > Thank you for the response clarifying some of my questions and concerns. Maybe binary classification might be more important than I thought as the authors mentioned. It would be better if strong justifications for the task were described in the paper (with better readability). To me, this paper is not yet good enough to pass the bar of ICLR. I recommend rejection.

---

### Official Review · Reviewer_WMWv · 2023-11-05

**Soundness:** 2 fair
**Presentation:** 2 fair
**Contribution:** 2 fair
**Rating:** 3
**Confidence:** 5

**Summary:**

The paper proposed a framework for binary image classification while to address two associated problems simultaneously: (i) making the resulting classifier adversarially robust, and (ii) get attribution maps so as to learn which regions are important for classification. To do this, the authors use a generative adversarial learning framework, where the generator maps an input image of one class into an image another by introducing minimal changes.  The discriminator is convert to behave like a classifier (damaged vs undamaged class) as well as a real/fake classifier. Upon training, the authors show that at test time, the learnt generator can be used get attribution map for an image, and that the resulting discriminator is a robust classifier. Results are shown on two binary classification tasks.

**Strengths:**

1. The authors make an interesting use case of the image to image translation framework where the convert an image from one class to another. Because of the nature of the dataset, where one class has some artifacts (damage) which the other class does not, the resulting generator appears to be *only* introducing the artifact when input is from undamaged class, and vice versa, *only* removing the artifact when the input is from the damaged class. Because of this, they can get very accurate attribution maps (in Fig. 2 and 4), which can almost be used to do a decent job at semantic segmentation (Fig. 5).

2. The authors have given a discussion for how their framework might be extended to multi-class classification setup.

**Weaknesses:**

1. It is not clear why the authors want to solve the two tasks simultaneously: trying to make a classifier adversarially robust is almost orthogonal to one wanting better attribution maps for that classifier. (these attribution maps should not even be called that technically, as I will explain later). There is not much motivation explaining this particular combination of problem. For example, some questions that the authors might want to consider and talk about is: is a classifier which has those abilities learnt simultaneously better than a classifier which has learnt them sequentially? Or is the robustness of the classifier presented in this work would have been better than a network which was *solely* trained to be adversarially robust? Right now the paper reads as if the authors randomly wanted to have a framework to solve two (seemingly) random problems.

2. The framework is not that easy to understand. In particular, it is not clear why the classification head is needed in the U-net of the generator. Overall, there seems to be many kinds of classifications happening at different stages. There is one happening in the generator, and then also in the discriminator. While the reader *can* follow along, the overall framework lacks a bit of intuition. This is also because the authors have claimed certain things in the text for which there is not much justification. For example, at the end of Section 3.2.2, the authors claim that the objective “bolsters both the training stability and the expressive capability of the generator”. It is not clear what expressivity exactly means, and how exactly are the authors measuring the stability of the generator training.

3. There is some confusion in the way results are presented. In Table 2, what is the difference between the top and bottom sections of the table? What do non-adversarially trained equivalents mean; i.e. how exactly were those generators and discriminators trained? Furthermore, the nomenclature used in the paper to refer to different models is a bit confusing across the paper. For example, in Section 4.5, the authors the phrase “D is more robust compared to its non-adversarial counterpart”. But the figure that they are referring to, Fig. 3, does not have any “D” in it. There is a “Hybrid D” and “D_fake”. I would strongly recommend the authors to be consistent with the naming scheme.

4. About the attribution maps: If I understand correctly, in Fig. 4, the way the authors are computing the saliency maps under the “Segmentation” column, which is their primary method, is through a difference between an input image ‘x’ and its transformed image G(x). However, the region highlighted by this difference does not mean it is the same region used by the discriminator to classify them as damaged or undamaged. In other words, just because we can see the difference between two kinds of images does not mean that the neural network is looking at the same kind of difference as well. Therefore, there is not much point of comparison to the methods in “Attribution methods”.

5. Since attribution maps will be anything that results in the image of one class to become like the image of the other class, the nature of the attribution map will depend on the *types* of classes in the dataset. The framework can learn the segmentation mask because the other class does not have that property. If the two classes were, for example, dogs and cars, then the attribution maps (the way the authors are obtaining them) will look very different, and will likely not be used for segmentation task. Therefore, the strongest point about the paper, which is the emergence of these saliency maps, is an outcome of this particular setting, and not a general phenomena.

**Questions:**

Comments:

1. The word damaged - in real-damaged vs real-undamaged is confusing. Maybe replace with a different word because damage might also mean adversarial example.
2. Eq. 1 loss formulation seems incorrect. Use the standard form of cross entropy.

---

> ### Author Response · Authors · 2023-11-22
>
> Thank you for your review. We appreciate the time and effort you have put into reading our work and communicating your assessment.
>
> We would like to address the points mentioned in "Weaknesses":
> 1. Recent studies demonstrate a converging relationship between adversarial robustness and model interpretability in neural networks. For instance, [1, 2] show that adversarially trained networks' loss gradients align with features that are intuitively salient to humans. These two works are mentioned in our related work section. \
> An advantage of our framework compared to a two-stage approach is efficiency: we create data augmentation, adversarial samples, and saliency maps in a single training run. Furthermore, the saliency maps in our method are actually a byproduct of our adversarial objective, which encourages the classifier to focus on perceptually relevant features. Using GANs for improving classifier robustness has been extensively studied (c.f. our related work section) - we made tweaks to those methods (including weight sharing between classifier and discriminator, regularization between original and generated image, etc.) which allow us to simultaneously leverage the adversarial examples as visual explanations. This is a clear benefit over current attribution methods and frameworks for adversarial robustness.
> \
> [1] Robustness May Be at Odds with Accuracy, Tsipras et al.\
> [2] Adversarial Explanations for Understanding Image Classification Decisions and Improved Neural Network Robustness, Woods et al.
>
> 2. We acknowledge two main points of complexity: (i) the generator’s classification head, and (ii) the role of cycle consistency loss. \
> (i) The generator's classification head is crucial as our generator is unconditioned, a notable deviation from similar models. This classification head infuses direct class information into the generator, enhancing its performance under the multi-task learning paradigm. While we were not able to include detailed ablation studies due to the page limit,  we plan to publish the codebase for reproducing our results, which may shed more light on the impact of different hyperparameters on the final model performance. \
> (ii) The cycle consistency loss, as discussed by Charachon et al. in “Leveraging conditional generative models in a general explanation framework of classifier decisions” (c.f. citation in Section 3.2.2), is proven effective in related contexts. It contributes to the overall stability and expressive capability of the generator, ensuring more reliable and coherent outputs.
>
> 3. The non-adversarially trained models in our study were developed using standard supervised learning techniques. Due to space constraints, we couldn't include all training details in the paper, but we intend to release our complete codebase post-review, which will offer comprehensive insights into our implementation. \
> The “D” in Section 4.5. refers to the “Hybrid D”, where “Hybrid” specifies the architecture of the discriminator used in this experiment. Following your suggestion, we also mentioned the discriminator’s architecture in the Section referring to the Figure in question. Also, we appreciate your feedback on consistency in naming. There was a wrong notation in the legend of Figure 3, where we noted D_fake instead of p_fake.  We have revised Figure 3 and its legend to align with the notation established in Section 3.2.1.
>
> 4. Given that our generator is unconditioned and receives updates through gradients from the discriminator, it's reasonable to infer that it minimally alters features critical to the discriminator's classification. The generator's primary guidance, apart from an auxiliary objective indicating the original image's class, comes from its goal to change the discriminator’s class and “fakeness” predictions. Therefore, the differences highlighted between 'x' and G(x) are indeed indicative of the regions the discriminator focuses on for its decision-making.
>
> 5. Our research primarily focuses on applications such as structural health monitoring or medical diagnosis. In these scenarios, the classes typically involve 'healthy' or 'undamaged' versus 'unhealthy' or 'damaged' states. In such contexts, our method's saliency maps are effectively comparable to segmentation labels. While our approach could extend to diverse class pairs like 'dogs' and 'cars,' the resulting saliency maps would differ and aren't intended for semantic segmentation. The aim of our study is to showcase the effectiveness of our method in highlighting regions relevant to the classification process, using cases where these regions align with segmentation maps for clear demonstration.
>
> We have also adapted the word damaged - in real-damaged vs real-undamaged - to a less ambiguous formulation. Regarding the cross entropy, we have double-checked our notation.

---

### Meta-Review · Area_Chair_Ynu7 · 2023-12-17

**Metareview:**

The paper addresses adversarial robustness and interpretability in image classifiers. The proposed framework is based on Generative Adversarial Networks (GANs), it generates counterfactual samples that enhance interpretability by highlighting salient regions and augment the dataset for increased robustness against adversarial attacks.

The reviewers raised several important concerns regarding
(1) presentation clarity and lack of motivation/justification behind simultaneously addressing adversarial robustness and improved attribution maps in the proposed framework – all reviewers, see e.g. Reviewer WMWv  and reviewer 9wk6 comments on how to improve;
(2) limited technical novelty in terms of the scope of binary classification – see all reviewer comments, and prior work (see Reviewer ujhc comment on ACGAN );
(3) the complexity of the framework, especially regarding the role of the classification head in the generator as opposed to making it conditional – see all reviewers comments, esp. Reviewer WMWv and Reviewer 9wk6 on how to improve; (4) unconvincing empirical evaluations that lack strong baseline comparisons, e.g. to generating semantic counterfactuals/adversaries  - see Reviewer 9wk6 and  Reviewer jyYD reviews. (5) Reviewer WMWv and reviewer Reviewer jyYD also challenge the interpretation of saliency maps, that the highlighted regions may not correspond to those used by the discriminator for classification, but may be specific to the dataset, and the fact that simply using absolute pixel differences of image translation as counterfactual explainability is not grounded.

The authors have addressed some of the concerns in their rebuttal. However (2)-(4) make it difficult to assess the benefits of the proposed approach and were viewed by AC as critical issues that need further revisions and another round of reviews. We hope the detailed reviews are useful for improving and revising the paper.

**Justification For Why Not Higher Score:**

All reviewers and AC are in consensus about rejection.

**Justification For Why Not Lower Score:**

N/A

---

### Decision · Program_Chairs · 2024-01-16

Reject